# RT-qPCR Expression Profiles of Selected Oncogenic and Oncosuppressor miRNAs in Formalin-Fixed, Paraffin-Embedded Canine Mammary Tumors

**DOI:** 10.3390/ani12212898

**Published:** 2022-10-22

**Authors:** Jessica Maria Abbate, Alessia Giannetto, Francesca Arfuso, Barbara Brunetti, Giovanni Lanteri

**Affiliations:** 1Department of Veterinary Sciences, University of Messina, Polo Universitario Annunziata, 98168 Messina, Italy; 2Department of Chemical, Biological, Pharmaceutical and Environmental Sciences, University of Messina, Polo Universitario Papardo, 98166 Messina, Italy; 3Department of Veterinary Medical Sciences, Alma Mater Studiorum University of Bologna, via Tolara di Sopra, Ozzano Emilia, 40064 Bologna, Italy

**Keywords:** canine mammary tumors, CMTs, microRNA, miRNA, RT-qPCR, biomarker, cancer, dog

## Abstract

**Simple Summary:**

Early diagnosis of mammary gland cancer would allow intervention in the early stage of the disease, with a better prognosis for canine patients. MiRNAs are short non-transcribed RNA molecules that can be involved in cancer as molecular regulators of tumor development and progression and recognized as early, cancer-specific biomarkers with diagnostic and prognostic value and promising targeted for cancer therapy. Our results demonstrate that MiRNAs are differentially expressed in canine mammary tumors (CMTs) compared with normal mammary gland. In particular, oncogenic miR-18a, miR-18b and miR-21 were significantly overexpressed in malignant tumors and their involvement in receptor-mediated carcinogenesis and proteoglycan remodeling, making them candidate biomarkers with a prognostic value in CMTs. We also found downregulation of the oncosuppressor miR-146b in both benign and malignant CMTs compared with the normal mammary gland. MiR-146b may regulate the production of pro-inflammatory cytokines with a crucial role in cancer development and is predicted to target genes involved in Toll-like receptor and MAPK-signaling pathways. By virtue of their possible involvement in neoplastic transformation and progression, investigated miRNAs may represent candidate biomarkers with prognostic relevance in dogs with mammary tumors.

**Abstract:**

MicroRNAs (miRNAs) can act as oncogenes or oncosuppressor genes, and their involvement in nearly all cancer-associated processes makes these small molecules promising diagnostic and prognostic biomarkers in cancer, as well as specific targets for cancer therapy. This study aimed to investigate the expression of 7 miRNAs (miR-18a, miR-18b, miR-22, miR-124, miR-145, miR-21, miR-146b) in formalin-fixed, paraffin-embedded canine mammary tumors (CMTs) by quantitative reverse transcription polymerase chain reaction (RT-qPCR). Twenty-six mammary samples were selected, including 22 CMTs (7 benign; 15 malignant) and 4 control samples (3 normal mammary gland and 1 case of lobular hyperplasia). Oncogenic miR-18a, miR-18b and miR-21 were significantly upregulated in malignant tumors compared with control tissues (*p* < 0.05). Conversely, oncosuppressor miR-146b was significantly downregulated in benign and malignant mammary tumors compared with control samples (*p* < 0.05) while, no group-related differences in the expression levels of miR-22, miR-124 and miR-145 were found (*p* > 0.05). Upregulated miRNAs found here, may regulate genes involved in receptor-mediated carcinogenesis and proteoglycan remodeling in cancer; while miRNA with reduced expression can regulate genes involved in Toll-like receptor and MAPK signaling pathways. According to the results obtained in the current study, the oncogenic and oncosuppressor miRNAs analyzed here are dysregulated in CMTs and the dysregulation of miRNA targets may lead to specific altered cellular processes and key pathways involved in carcinogenesis. Of note, since oncogenic miRNAs predicted to regulate neoplastic cell proliferation and hormonal activities, they may play an active role in neoplastic transformation and/or progression, having mechanistic and prognostic relevance in CMTs.

## 1. Introduction

Mammary tumors are the second most commonly diagnosed type of cancer in dogs, accounting for up to 70% of all tumors in hormonally intact females [1,2], thus representing a significant clinical problem. Almost half of canine mammary tumors (CMTs) are malignant [1,3] and 50% of carcinomas metastasizes to local lymph nodes, lungs, and occasionally invades bones [4,5], with high mortality rate (i.e., 20–55%) if dogs are not treated on time [6,7,8]. Histological examination still remains the gold standard method for accurate classification and grading of CMTs [9], and so far, invasive surgery is considered the only therapeutic treatment with a high success rate. Indeed, chemotherapy has not been shown to be as effective in treating CMTs as in human breast cancer [10,11], and standard chemotherapeutics do not significantly improve overall survival time [12]. Therefore, due to aggressive biological behaviour and limited therapeutic opportunities, early diagnosis of CMTs would allow intervention in the early stage of the disease, improving the overall survival time of canine patients. 

MicroRNAs are short (18–22 nucleotides), non-transcribed, RNA molecules in cells that bind specific sequences in messenger RNA (mRNA) strands and direct gene silencing, contributing to a complex called “RNA-silencing complex” [13,14]. MiRNAs can regulate up to 30% of genes via post-transcriptional control in target cells and, of note, a single miRNA can target hundreds of different mRNAs, affecting a multitude of transcripts [15]. Noteworthy, miRNAs regulate virtually all cellular processes in both health and disease and play a key role in cancer-associated cellular processes [13]. Dysregulation of miRNA expression profiles is a common occurrence in cancer due to epigenetic modifications, transcription alteration or as a result of DNA copy number aberrations [16]. Consequently, the genetic instability of tumors leads to changes in miRNA-controlled gene regulatory networks [16,17,18] and promotes cancer-related signaling pathways [15]. MiRNAs can act as oncogenes (oncomiRs) and as tumor suppressor genes, and each group exerts its effect depending on its level of expression [16]. Commonly, miRNAs that negatively regulate the oncoproteins are downregulated in cancer, while miRNAs that negatively regulate tumor suppressor genes are upregulated [19]. Due to their role in tumorigenesis and their abundance in tumor microenvironment [20,21,22,23], miRNAs have been extensively studied as potential cellular and molecular biomarkers for prediction, diagnosis, progression monitoring, prognosis determination and as therapeutic targets for various cancers [24,25]. They act as tissue biomarkers, and free miRNAs are also available in most biological fluids, packaged into extracellular vesicles (i.e., exosomes), enwrapped with microvesicles during their biogenesis, and/or binding to proteins, protecting them from endogenous RNases in the extracellular environment [26], also making their sampling minimally invasive. 

Several non-exosomal and exosome-derived miRNAs are already known to be dysregulated in canine mammary gland cancer, and noteworthy, high similarities in terms of miRNA dynamics and functions have been found between canine mammary and human breast cancer [19,27]. MiRNAs were investigated in primary and metastatic CMT tissues to find candidate biomarkers in distinguishing between different stages of CMTs [28]. Notably, miR-21 was found significantly upregulated in mammary carcinomas and, miR-29b, miR-101, miR-125a, miR-143 and miR-145 differed significantly in metastatic cells compared with primary cancer cells, making them potential biomarkers of metastasis [28]. More recently, a total of 124 miRNAs were found to differ significantly between metastatic and non-metastatic CMTs by RT-qPCR and RNA hybridization arrays and included in a miRNA signature that predicts shorter survival [29]. MiR-141 has been shown to be upregulated in several CMT cell-lines by RT-qPCR and experimentally validated to downregulate tumor suppressor p16/INK4a [30]. Finally, upregulated miR-19b, miR-18a and miR-124 have been found in serum from dogs with mammary carcinoma and suggested as candidate biomarkers for prognosis [31,32].

Dysregulation of miRNAs expression and their involvement in tumorigenesis have made them interesting biomarkers in cancer research and, miRNAs evaluation would represent a tool to find accurate biomarkers with diagnostic and prognostic value in CMTs and to discover miRNA-based drugs as promising targeted therapies to be used in the near future. Nowadays, investigations focusing on miRNA-based biomarkers have been conducted in canine mammary cancer using surgically removed CMT tissues [19,28,29,33,34], tumor cell lines [30,35,36] and serum/plasma samples [31,37], and data obtained are sometimes conflicting depending on the RNA sources and miRNA profiling and normalization techniques. Furthermore, bioinformatic analysis to identify miRNA-controlled target genes and miRNA-regulated cancer signaling pathways are performed only occasionally. In view of the above considerations, this study aims to investigate the expression levels of selected oncogenic and oncosuppressor miRNAs (miR-18a; miR-18b; miR-22; miR-124; miR-145; miR-21, miR-146b) supposed to be involved in the pathogenesis of canine mammary cancer by quantitative reverse transcription polymerase chain reaction (RT-qPCR), and to identify target genes and specific pathways regulated by differently expressed miRNAs. Furthermore, to the Authors’ knowledge, this is the first study that reports miRNAs expression in formalin-fixed, paraffin-embedded (FFPE) canine mammary tumors. 

## 2. Materials and Methods

### 2.1. Mammary Samples

Canine mammary tumors were selected among the samples referred for diagnostic purposes between 2021 and 2022, identified in the archive of the AniCura Veterinary Hospital “I Portoni Rossi,” Bologna, Italy. Histological examination was performed by a board-certified pathologist (B.B.) on three-μm thick tissue sections stained with haematoxylin-eosin (HE). Classification of CMTs was performed according to a recent classification published by the Davis-Thompson DVM Foundation [38]. Histological grading was performed according to Peña et al. [6] and based on histological features of tubules formation, nuclear pleomorphism and mitotic count. Normal mammary gland tissue samples were collected from dogs undergoing postmortem examination and used as negative controls for molecular investigations. 

### 2.2. RNA Extraction and Reverse Transcription Reaction

Two sections, 10–15-μm thick, of a 2-mm diameter selected area for each sample block, representative of tumors and normal mammary gland and identified on HE-stained histological slides, were sampled using a disposable biopsy punch and collected in a sterile 1.5 mL microcentrifuge tube. Purification of total RNA from FFPE tissue sections was performed using miRNeasy^®^ FFPE Kit (QIAGEN, Milan, Italy; cat. no. 217504) and Deparaffinization Solution (QIAGEN, Milan, Italy; cat. no. 19093) according to the manufacturer’s instructions. 

Reverse transcription (RT) reaction was performed using miRCURY^®^ LNA^®^ RT Kit (QIAGEN, Milan, Italy; cat. no. 339340) and provided UniSp6 RNA Spike-in was used as an internal quality control for the cDNA synthesis. Reverse transcription reaction was set up in 10-μL total reaction volume containing 2 μL of 5× miRCURY RT Reaction Buffer, 4.5 μL of RNase-free water, 1 μL of 10× miRCURY RT Enzyme Mix, 0.5 μL of synthetic RNA spike-in. Finally, 20 ng of extracted RNA in a 2 μL total volume was added to each RT reaction tube. Obtained cDNA was stored at −20 °C for further analysis. Concentration of nucleic acids (RNA; cDNA) were measured using the Nanodrop Spectrophotometer (NanoPhotometer N50, IMPLEN, USA).

### 2.3. Selected miRNAs and Real-Time Quantitative PCR (RT-qPCR)

The following miRCURY^®^ LNA^®^ miRNA PCR Assays (QIAGEN, Milan, Italy; Cat. no. 339306) were selected: gga-miR-18a-5p (assay ID-YP02100185); gga-miR-18b-5p (assay ID-YP02100265); hsa-miR-22-3p (assay ID–YP00204606); cbr-miR-124 (assay ID–YP02103368); has-miR-145-5p (assay ID–YP00204483); has-miR-21-5p (assay ID–YP00204230); has-miR-146b-5p (assay ID–YP02119310). Assays used as reference controls for miRNA normalization included: hsa-miR-16-5p (assay ID–YP00205702) and hsa-let-7a-5p (assay ID–YP00205727) [29,36]. 

RT-qPCR was performed using miRCURY LNA SYBR Green PCR Kit (QIAGEN, cat. no. 339345, 339346) following the manufacturer’s instructions. Quantitative reaction was performed in duplicate per each sample in a 20 μL total reaction volume, containing 10 μL of 2× miRCURY SYBR Green Master Mix, 2 μL of specific miRNA assay, 2 μL of RNase-free water and 6 μL of cDNA (60× diluted), and performed using a Rotor-Gene^®^ Real-Time PCR system (QIAGEN). The cycling conditions were set as follows: 95 °C for 2 min, 40 cycles of 95 °C for 10 s and 56 °C for 60 s. MiRNA expression levels were presented in terms of fold change normalized to endogenous controls using the formula $\Delta \Delta C_{q}$.

### 2.4. Gene Target Predictions, Gene Ontology and KEGG Pathway Enrichment

Predicted target genes for each miRNA were generated using the miRDB online resource and analysis platform (http://mirdb.org (accessed on 30 August 2022) and all targets having scores greater than 80 were selected, representing the most confident gene predictions [39]. All target genes obtained in the miRDB were uploaded into the DAVID database (https://david.ncifcrf.gov (accessed on 2 September 2022) using official gene symbols, and enriched gene ontology terms and KEGG pathways were identified [40,41,42]. 

### 2.5. Statistical Analysis

All data were tested for normal distribution using the Kolmogorov-Smirnov test. Since data resulted in abnormal distribution (*p* < 0.05), a nonparametric statistical analysis was applied. In particular, the Kruskal-Wallis test was performed to assess statistically significant differences among samples, followed by Dunn’s multiple comparison test. Statistical analyses were performed using GraphPad Prism version 4.00 (GraphPad Software, San Diego, CA, USA, 2003). The DAVID software provided raw and Benjamini-corrected *p*-values, and a crude threshold of *p* < 0.06 was selected for enriched KEGG pathways [43]. 

## 3. Results

### 3.1. Mammary Samples and Histological Classification of CMTs

A total of 26 mammary samples from dogs with a median age of 10 years (range: 4–12) were used in this study. In particular, 22 CMTs were used: 7 benign tumors (Group B) and 15 malignant tumors (Group M). Malignant tumors were: grade I, 8 samples (Group M1); grade II, 2 samples (Group M2); grade III, 5 samples (Group M3). Three samples of normal mammary gland and 1 sample of lobular hyperplasia were used as negative control cases (Group C). Histological classification and grading for CMTs is reported in Table 1. 

### 3.2. miRNAs Expression Levels 

Selected miRNAs were detected in all samples. Statistical analysis of data revealed a significant effect of group (*p* < 0.05) on the expression levels of miR-18a, miR-18b, miR-21 and miR-146b, whereas no group-related differences in the expression levels of miR-22, miR-124 and miR-145 were found (*p* > 0.05) (Figure 1 and Figure 2). 

MiR-18a, miR-18b and miR-21 were upregulated in malignant tumors compared with the normal/hyperplastic mammary gland; furthermore, the expression level of both miR-18a and miR-21 was upregulated in malignant neoplasms compared with benign tumors. Conversely, miR-146b was significantly downregulated in both benign and malignant mammary tumors compared with control cases, and significant differences were also observed in grade II malignant tumors compared with both grade I and III, and benign tumors (Figure 2). 

### 3.3. MicroRNAs Targets, Functional Annotation and Pathway Enrichment of All miRNAs

Target genes of statistically significant up- or downregulated miRNAs were predicted using miRDB online resource. The number of totals predicts identified per each miRNA ranged from 205 to 780 genes, and predicted genes having a target score greater than 80 were considered, including 45 target genes for both miR-18a and miR-18b, 69 for miR-21 and 47 for miR-146b. Detailed list of predicted target genes for significant up- and downregulated miRNAs is reported in Appendix A. 

Gene functional annotation and pathway enrichment analysis was performed using the DAVID bioinformatic tool. The complete list of target genes (*n* = 158) obtained in miRDB, was submitted in the DAVID database using official gene symbols and, 156 out of 158 uploaded targets were successfully mapped to canine genes, whereas two gene symbols (i.e., LOC483164; INSYN2) were identified as unknown. The gene functional annotation was carried out organizing the annotation types in three categories: biological process, cellular component, and molecular function. 

Biological processes included genes mainly involved in the regulation of transcriptional activity and protein binding, and enriched biological term included the “Ubiquitin conjugation pathway” (*p*-value = 0.017) associated with genes FBXL3, TRAF6, KLHL15, PELI1, RMND5A, SOCS5. Cellular component items focused on nucleus (*p*-value = 0.019) with the involvement of the following genes: BBX, ELF2, ESF1, GATAD2B, HORMAD1, RORA, TRAF6, APPL1, CREBL2, ESR1, HSF5, HNRNPD, IRF2, KLHL15, MAGOH, NFIA, NFIB, PITX2, PPARA, PBRM1, UIMC1. 

Molecular processes mainly involved genes implicated in DNA binding activity and regulation of transcription. Enriched molecular functions included: DNA-binding (*p*-value = 0.0017), associated with genes ELF2, RORA, THAP5, ESR1, HSF5, IRF2, NFIA, NFIB, PITX2, PPARA, STRBP, ZNF287; the function “Activator” (*p*-value = 0.027) associated with genes HIF1A, IRF2, NFIA, NFIB, PPARA. 

Enriched pathways obtained from KEGG pathway enrichment analysis using DAVID Bioinformatics Resource are reported in Table 2. 

The graphic representation of the enriched “Pathways in cancer” involving 10 targets genes for oncogenic and oncosuppressor miRNAs is reported in Figure 3. Pathway information is generated by KEGG using DAVID Bioinformatic tool (https://david.ncifcrf.gov (accessed on 2 September 2022).

### 3.4. Gene Ontology and Pathway Enrichment Differentiating between Up- and Downregulated miRNAs

A total of 112 predicts obtained for upregulated miR-18a, miR-18b and miR-21, were uploaded in the DAVID database, and 111 out of 112 were successfully mapped. The target genes controlled by the upregulated miRNAs were significantly enriched in the biological processes of “Ubiquitin conjugation pathway” (*p*-value = 0.021), associated with FBXL3, KLHL15, PELI1, RMND5A, SOCS5 genes; and “Transcription regulation” (*p*-value = 0.043) associated with genes ELF2, GATAD2B, ESR1, IRF2, PPARA, PBRM1, ZNF287. Enriched cellular components focused on the nucleus (*p*-value = 0.0087), associated with genes BBX, ELF2, ESF1, GATAD2B, RORA, CREBL2, ESR1, HSF5, IRF2, KLHL15, MAGOH, NFIA, NFIB, PITX2, PPARA, PBRM1, UIMC1. Enriched molecular terms included: DNA-binding (*p*-value = 0.00041) associated with ELF2, RORA, THAP5, ESR1, HSF5, IRF2, NFIA, NFIB, PITX2, PPARA, ZNF287; “Activator” (*p*-value= 0.009) associated with genes HIF1A, IRF2, NFIA, NFIB, PPARA. 

Enriched pathways for upregulated miRNAs included: “Chemical carcinogenesis–receptor activation” (*p*-value = 0.018) associated with ESR1, FGF18, JAG1, KPNA6, PPARA genes; the pathway “Proteoglycans in cancer” (*p*-value = 0.089) associated with genes ESR1, FRS2, HIF1A, PDCD4. 

Analysis of downregulated miR-146b was based on 46 successfully mapped predicted targets. Enriched biological terms included “Spermatogenesis- Differentiation” (*p*-value= 0.049) associated with genes HORMAD1 and STRBP. No enriched molecular functions and cellular components were found. Enriched pathways included: “Toll-like receptor signaling pathway” (*p*-value = 0.027) associated with genes CD80, TRAF6 and IRAK1; “MAPK signaling pathway” (*p*-value = 0.038) associated with genes MET, TRAF6, ERBB4 and IRAK1. 

## 4. Discussion

MicroRNAs have functional roles in various diseases and in cancer, where they can act as oncogenes or tumor suppressor genes [16]. Differentially expressed miRNAs have been implicated in neoplastic conditions in dogs as molecular regulators of tumor development and progression, and circulatory miRNAs have been recognized as specific biomarkers for individual tumor type, including mammary gland tumors [28,29,30]. In the present study, the expression levels of 7 miRNAs presumably involved in the pathogenesis of CMTs were investigated and, to the best of the Authors’ knowledge, this is the first study that reports miRNAs expression level using FFPE mammary samples. The results obtained here suggest that the selected miRNAs are differentially expressed in neoplastic samples compared with the nonneoplastic mammary gland, with significantly up- and down-regulated miRNAs that could represent potential diagnostic and prognostic biomarkers, as well as promising targeted therapy in canine patients with mammary cancer. In particular, oncogenic miR-18a, miR-18b and miR-21 were significantly upregulated in mammary tumors compared with the normal mammary gland; whereas, the progressive decrease in the expression of oncosuppressor miR-146b was significantly related to malignancy. 

Upregulation of both miR-18a and miR-18b in malignant tumors found here largely corroborates previous studies on miRNAs in CMTs [30,31,36]. In particular, miR-18a was found significantly overexpressed in malignant canine mammary epithelial cell lines and in their exosomes [30,36] as well as a significantly higher serum concentration of miR-18a was found in dogs with mammary carcinomas and histological evidence of lymphatic invasion, representing a possible candidate biomarker for prognosis [31]. Based on gene target prediction analysis, both miR-18a and miR-18b show a high probability of targeting to 3′ UTR of the estrogen receptor ESR1 (score 99 in miRDB) mRNA. Furthermore, miR-22 investigated in this study was also predicted to target ESR1 mRNA (miRDB score 84). However, unlike other studies [36], expression level of miR-22 in our study did not differ significantly between CMTs and nonneoplastic conditions. It has long been recognized that human and canine mammary neoplasms lose estrogen receptor (ER) expression along with increasing tumor grade [44,45], and a high target score for ESR1 mRNA for both miR-18a and miR-18b, together with their overexpression in malignant cases, indicate that specific miRNAs such as miR-18a and miR-18b may contribute to this loss of hormone receptor activity, providing a potential molecular mechanism for the loss of ER with increased grade of malignancy. Therefore, both miRNAs, may represent non-invasive biomarkers of hormonal status and phenotype with a prognostic value in canine mammary tumors. 

Significant upregulation of miR-21 was found in malignant mammary cases compared with both benign tumors and control cases in our study. It has been well established that miR-21 overexpression is considered a hallmark of carcinogenesis in CMTs [19,46]. In particular, miR-21 selectively targets tumor suppressor genes and, acting as oncomiR through the inhibition of apoptosis in neoplastic cells [19,46]. Furthermore, in human breast tumors, miR-21 has been shown to simultaneously downregulate multiple metastasis-related tumor suppressor genes, including programmed cell death 4 (PDCD4), encouraging cell invasion and metastasis [46]. Although with a very low target score, gene target analysis performed in our study, included PDCD4 mRNA as a predicted target of miR-21 (score 81 in miRDB). Conversely, different from human breast cancer, miR-21 had a very high likelihood of targeting to 3′ UTR of the fibroblast growth factor 18 FGF18 (score 97 in miRDB) mRNA. In breast cancer, FGF18 promotes cell proliferation through the ERK/c-Myc signaling pathway and stimulates the production of epithelial-to-mesenchymal transition (EMT) factors, thus, promoting neoplastic cell migration and invasion [47]. Noteworthy, in the current study, the highest level of miR-21 expression was observed in the two malignant tumors with evidence of neoplastic emboli in lymphatic vessels (i.e., tubular carcinoma, grade III; inflammatory carcinoma, grade III) (Data not shown). In primary breast cancer, miR-21 has been included in a miRNA signature that predicts tumor recurrence and shorter survival [48], and, overexpression was positively correlated with metastatic carcinomas and poor prognosis, make it a good metastatic and prognostic biomarker in canine patients [28]. 

The Gene Ontology and pathway analysis in our study suggest that upregulated miRNAs are engaged in transcription regulation among enriched biological processes, inducing receptor activation and modulating expression patterns of proteoglycans and their biosynthetic activities. Therefore, upregulation of miRNAs target genes leads to deregulation of specific cellular and molecular processes probably involved in the pathogenesis of CMTs. Receptor-mediated carcinogenesis is a major mechanism of cancer induction, and in particular estrogen signaling pathway has been associated with mammary carcinogenesis [1,49]. Long-term exposure to the hormone estrogen and its binding to the estrogen receptor positively regulates the transcription of more proliferative genes and suppresses apoptotic genes, leading to the proliferation of cells with DNA mutations [49]. Regarding proteoglycans (PGs), they are the main biomolecules of the extracellular matrix and recent evidence shows that they represent an important target of microRNAs, and dysregulated expression of specific microRNAs results in aberrant expression patterns of proteoglycans and their biosynthetic activities [50]. Especially in malignant tumors, extensive remodeling of the tumor stroma is associated with marked alterations in proteoglycan expression and structural variability, which in turn affects tumor cell proliferation, migration and epithelial-to-mesenchymal transition [50]. These macromolecules mainly contribute to the formation of a temporary permissive matrix for tumor growth, influencing tissue organization, cell-cell and cell-matrix interactions and cell signaling, also regulating tumor stroma angiogenesis and the development of drug resistance [51]. 

Downregulation of miR-146b was identified in CMTs compared with nonneoplastic mammary tissues in the present study. In breast cancer cell, miR-146b seems to act as tumor suppressor genes [52], while, to the best of the Authors’ knowledge no studies investigate the role of this miRNA in canine mammary cancer, to date. In human breast cancer cells, miR-146b overexpression significantly inhibits cell proliferation and EMT by upregulating E-cadherin and downregulating vimentin expression, modulating the transcription of the EMT-related genes [52]. Furthermore, miR-146b inhibits NF-κB-mediated production of IL-6, a proinflammatory cytokine associated with cancer, STAT3 activity, and IL-6/STAT3-driven breast cancer cells migration and invasion [53]. Of note, as the gene encoding miR-146b is decreased in neoplastic cells, therapies that reintroduce or stimulate this miRNA have been suggested as beneficial to patients [53].

Target prediction analysis for miR-146b in our study suggests interleukin 1 receptor associated kinase 1 IRAK1 mRNA as the most predicted (miRDB target score 99). In triple-negative breast cancer, IRAK-1 regulates the expression of inflammatory genes by immune cells [54]. Since inflammatory cytokines interleukin 1 (IL-1) and interleukin 6 (IL-6) play an important role in the regulation of cancer development [55], the role of miR-146b in regulating cytokines production in CMTs is worthy of future investigations. Interestingly, enriched pathways for miR-146b included “Toll-like receptor signaling pathway” and Toll-like receptors (TLRs), modulate activation/ inhibition of immune and non-immune cells, primarily affecting the production of pro-inflammatory cytokines. Especially, TLR4 protein was found to be involved in breast cancer development, and its overexpression negatively correlates with prognosis [56]. Finally, abnormally activated MAPK-(mitogen-activated protein kinase) signaling pathway is a common finding in different type of cancer, including breast cancer, being involved in proliferation, migration and invasion characteristics of neoplastic cells [57,58]. 

## 5. Conclusions

Oncogenic miR-18a, miR-18b, miR-21, and oncosuppressor miR-146b are dysregulated in FFPE canine mammary tumors and, dysregulation of miRNAs targets leads to altered specific cellular processes and key pathways involved in carcinogenesis. Since investigated oncogenic miRNAs predicted to regulate neoplastic cell proliferation and hormone activities, they could play an active role in neoplastic transformation and/or progression of CMTs, having mechanistic and prognostic relevance. Among early biomarkers of cancer, miRNAs evaluation has been shown to be a very promising method for diagnosis and prognosis, as well as to discover miRNA-based drugs as targeted therapies. Further studies are needed to accurately identify targets genes and to confirm mechanistic and prognostic relevance of oncogenic miRNAs analyzed here, to be considered as valuable candidate biomarkers for canine patients ‘prognosis in the near future. Furthermore, the potential beneficial role of miR-146b also in dogs with mammary tumors is worthy of further investigations.

## Figures and Tables

**Figure 1 animals-12-02898-f001:**
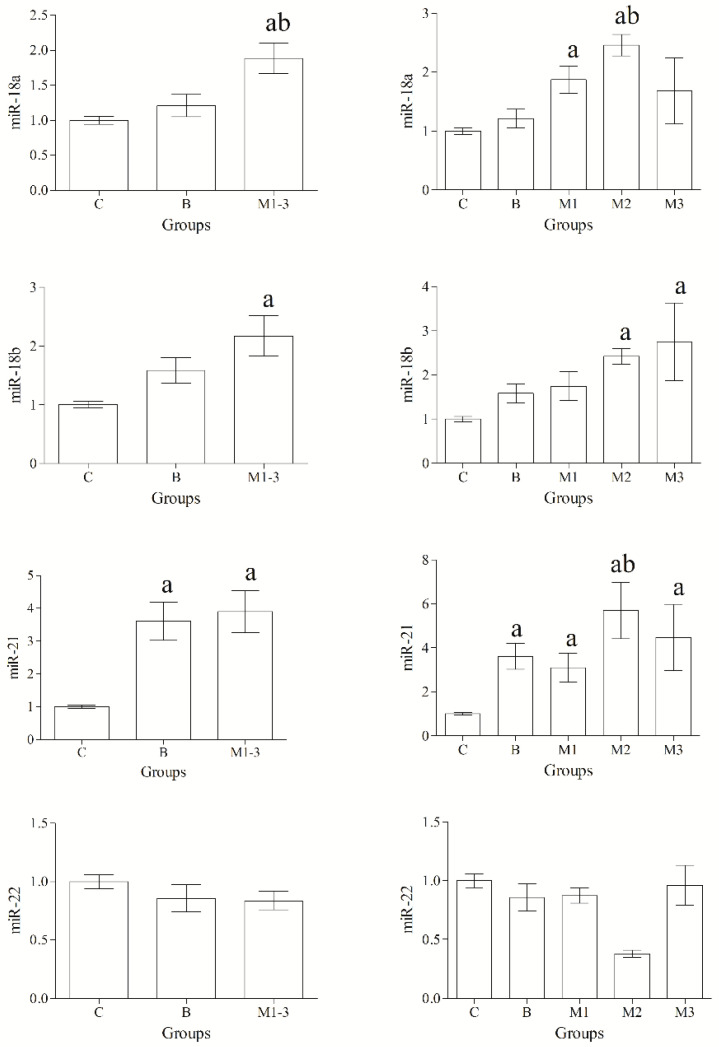
Expression levels of miR-18a, miR-18b, miR-21, miR-22. MiR-18a, miR-18b and miR-21 were upregulated in malignant tumors (M1-3) compared with the control group (C). The expression level of miR-18a and miR-21 was significantly upregulated in malignant neoplasms compared with benign tumors (B). Significances (*p* < 0.05): ^a^ vs. Group C, ^b^ vs. Group B.

**Figure 2 animals-12-02898-f002:**
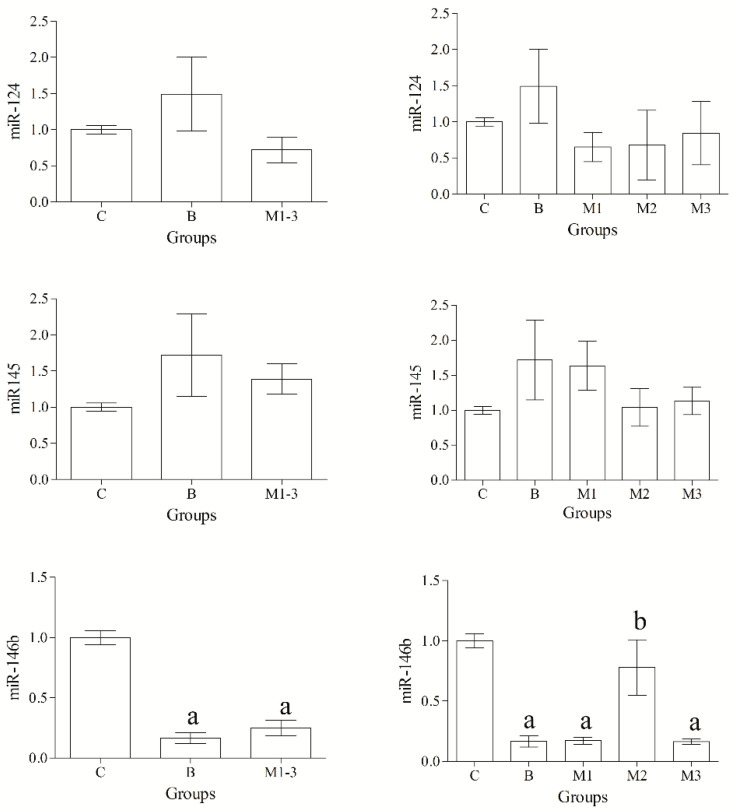
Expression levels of miR-124, miR-145, miR-146b. MiR-146b expression was significantly downregulated in benign (B) and malignant (M1-3) tumors compared with control mammary gland (C). Significant upregulation for miR-146b expression was observed in malignant tumors grade II (M2) compared with grade I (M1) and III (M3) and benign tumors (B). Significances (*p* < 0.05): ^a^ vs. Group C; ^b^ vs. Group B, M1 and M3.

**Figure 3 animals-12-02898-f003:**
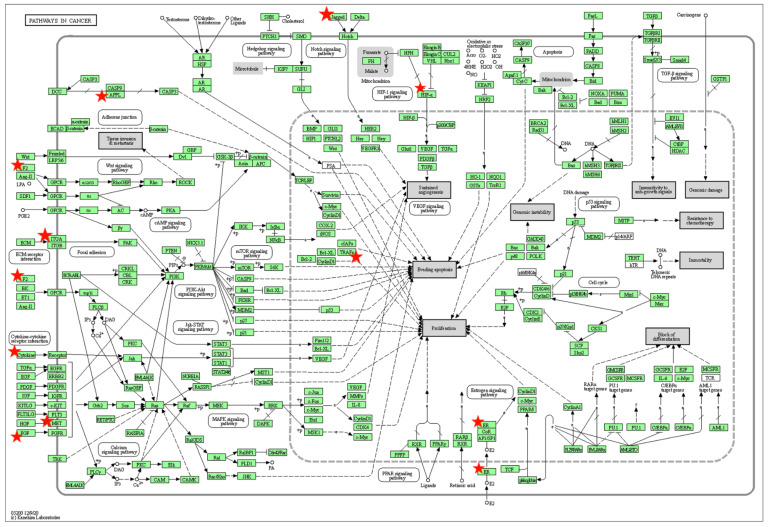
Enriched “Pathways in cancer” in Canine Mammary Tumors. Predicted target genes for oncogenic and oncosuppressor miRNAs are marked with red star, including 5 target genes for miR-146b (MET; TRAF6; APPL1; F2; IL12A); 3 predicted genes for both miR-18a and miR-18b (ESR1; HIF1A; ITGA6) and 2 predicts for miR-21 (FGF18; JAG1).

**Table 1 animals-12-02898-t001:** Histological Classification and Grading for Canine Mammary Tumors.

Tumor Type	Histological Classification	No.	Grade of Malignancy	Lymphatic Invasion
Benign Tumor (*n* = 7)	Tubular Adenoma	1		
	Tubulopapillary Adenoma	1		
	Complex Adenoma	2		
	Intraductal Papillary Adenoma	2		
	Benign Mixed Tumor	1		
Malignant Tumor (*n* = 15)	Tubular Carcinoma	1 1	I III	Yes
	Tubulopapillary Carcinoma	1	I	
	Complex Carcinoma	1 1 1	I I I	
	Intraductal Papillary Carcinoma	1 1	I I	
	Mixed Carcinoma	1 1	I II	
	Solid Carcinoma	1 1 1 1	II III III III	
	Inflammatory Carcinoma	1	III	Yes

**Table 2 animals-12-02898-t002:** Pathway Enrichment for all statistically significant expressed miRNAs.

KEGG Pathway	Genes	Number of Genes (%)	*p*-Value	Fold Enrichment	Corrected *p*-Value
Neurotrophin signaling pathway	TRAF6, FRS2, IRAK1, MAP3K1, NTF3	5 (3.2)	0.014	5.3	0.82
Alcoholic liver disease	TRAF6, CPT1C, IRAK1, IL12A, PPARA	5 (3.2)	0.020	4.7	0.82
Proteoglycans in cancer	MET, ERBB4, ESR1, FRS2, HIF1A, PDCD4	6 (3.8)	0.023	3.6	0.82
Pathways in cancer	MET, TRAF6, APPL1, F2, ESR1, FGF18, HIF1A, ITGA6, IL12A, JAG1	10 (6.4)	0.024	2.3	0.82
MAPK signaling pathway	MET, TRAF6, ERBB4, FGF18, IRAK1, MAP3K1, NTF3	7 (4.5)	0.025	3.0	0.82
Toll-like receptor signaling pathway	CD80, TRAF6, IRAK1, IL12A	4 (2.6)	0.041	5.1	1.0
Toxoplasmosis	TRAF6, ITGA6, IRAK1, IL12A	4 (2.6)	0.052	4.7	1.0
Chemical carcinogenesis–receptor activation	ESR1, FGF18, JAG1, KPNA6, PPARA	5 (3.2)	0.070	3.2	1.0
RIG-I-like receptor signaling pathway	TRAF6, IL12A, MAP3K1	3 (1.9)	0.093	5.8	1.0

## Data Availability

Not applicable.

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
