# Peer review of "RT-qPCR Expression Profiles of Selected Oncogenic and Oncosuppressor miRNAs in Formalin-Fixed, Paraffin-Embedded Canine Mammary Tumors"

_animals, 2022, doi:10.3390/ani12212898_

Round 1

Reviewer 1 Report

The manuscript "RT-qPCR Expression Profiles of selected Oncogenic and Oncosuppressor miRNAs in Formalin-fixed, Paraffin-embedded Canine Mammary Tumors" is well written and executed. I have some observations:

- Figures 1 and 2: images quality of histograms should be improved

- The legend to Figures 1 and 2 should provide more detail. The explanation of the statistical significance must be included there

- The study considered only 2 cases of histological grade II carcinomas, a low sample size. It is suggest to augment these cases and reanalyze the expression results, or discuss this point

- line 248-249: delete this phrase because is very general 

- the information displayed in table 2 and figure 3 is general and does not detail the miRNAs with changes in expression between benign and malignant tumors. It is recommended to specify so that this analysis is more understandable

Author Response

Dear Reviewer,

Thank you very much for reviewing the manuscript entitled “RT-qPCR Expression Profiles of selected Oncogenic and Oncosuppressor miRNAs in Formalin-fixed, Paraffin-embedded Canine Mammary Tumors”.

We have addressed all of your concerns and detailed responses to your comments are provided in the attachment.

Reviewer 2 Report

Dear Authors,

This study aims to investigate the expression levels of some selected oncogenic and oncosuppressor miRNAs that supposed to be involved in the pathogenesis of canine mammary cancer. This investigation can allow the detection of potential target genes and pathways involved in development and progression of CMT, and thus the identification of potential prognostic and therapeutic biomarkers.

I found this study relevant and interesting, but I have some comments:

- For a descriptive study without prognosis, it included very few samples, which makes important to reinforce these results with other futures studies, with more data.

- It would be very interesting to evaluate possible associations between the microRNAs levels with other histopathologic features, like the presence of lymphatic invasion by neoplastic cells or the presence of regional metastasis (once usually the inguinal lymph node is also removed during the surgery and use for stage the disease (TNM).

Finally, there is some specific comments of the text:

- line 14: “MiRNAs are short non-transcribed RNA molecules involved in cancer” – I think that is more correct if you put “that can be involved in cancer”

- 136-137: “Two sections (…) of a 2mm diameter selected area for each sample block, (…), were sampled and collected”

How was the sample selected? Which criteria was used? How was the tissue removed from the paraffin block? Please add that information.

- Figure 1 and 2:

What is the C group? It is the control group? Is not define in the figure nor in the text.

The graphs are difficult to see when the article is printed– Please increase the size.

- line 249:

Please define which are the neoneoplastic mammary tissue cases?

- line 254: “significant differences were also observed…”

Please detail what are those differences (or indicate the figure where we can see them)

- Figure 3: Like the Figure 1 or 2 – it is difficult to see the image when we print the article. Can you increase the size?

Author Response

(The authors gave the same response as above.)
